# Effects of crystalline lens rise and anterior chamber parameters on vault after implantable collamer lens placement

**Quan Zou, Sen Zhao, Lei Cheng, Chao Song, Ping Yuan, Ran Zhu** *

Department of Center of Refraction, The Affiliated Xuzhou Municipal Hospital of Xuzhou Medicine University, Xuzhou, Jiangsu Province, P. R. China

* 630yifan@163.com

**Data Availability Statement:** Data are available from the Xvzhou Hospital Institutional Data Access / Ethics Committee of Xvzhou Hospital for researchers who meet the criteria for access to

## Abstract

### Background

To analyze vault effects of crystalline lens rise (CLR) and anterior chamber parameters (recorded by Pentacam) in highly myopic patients receiving implantable collamer lenses (ICLs), which may avoid subsequent complications such as glaucoma and cataract caused by the abnormal vault.

### Methods

We collected clinical data of 137 patients with highly myopic vision, who were all subsequent recipients of V4c ICLs between June 2020 and January 2021. Horizontal ciliary sulcus-to-sulcus diameter (hSTS) and CLR were measured by ultrasonic biomicroscopy (UBM), and a Pentacam anterior segment analyzer was used to measure horizontal white-to-white diameter (hWTW), anterior chamber depth (ACD), anterior chamber angle (ACA), anterior chamber volume (ACV), CLR, and postoperative vault (Year 1 and Month 1). The lens thickness (LT) was determined by optical biometry (IOL Master instrument). The predictive model was generated through multiple linear regression analyses of influential factors, such as hSTS, CLR, hWTW, ACD, ACA, ACV, ICL size, and LT. The predictive performance of the multivariate model on vault after ICL was assessed using the receiver operating characteristic (ROC) curve with area under the curve (AUC) as well as the point of tangency.

### Results

Average CLR assessed by UBM was lower than the average value obtained by Pentacam (0.561 vs. 0.683). Bland-Altman analysis showed a good consistency in the two measurement methods and substantial correlation (r = 0.316; $P$ = 0.000). The ROC curve of Model 1 (postoperative Year 1) displayed an AUC of 0.847 (95% confidence interval [CI]: 74.19–95.27), with optimal threshold of 0.581 (sensitivity, 0.857; specificity, 0.724). In addition, respective values for Model 2 (postoperative Month 1) were 0.783 (95% CI: 64.94–91.64) and 0.522 (sensitivity, 0.917; specificity, 0.605).

confidential data. The study data are only available upon request, because they are owned by The Affiliated Xuzhou Municipal Hospital of Xuzhou Medicine University, and the Research Ethics Committee or Institutional Review Board has imposed them. The email address for the ethics committee is xyykjc@163.com.

**Funding:** This work was supported by Funding: Postgraduate Research & Practice Innovation Program of Jiangsu Province (SJCX22-1263). The funders had no role in study design, data collection and analysis, decision to publish, or preparation of the manuscript.

**Competing interests:** The authors have declared that no competing interests exist.

## Conclusion

CLR and anterior chamber parameters are important determinants of postoperative vault after ICL placement. The multivariate regression model we constructed may serve in large part as a predictive gauge, effectively avoid postoperative complication.

## Introduction

Throughout the world, ametropia is increasing year after year and is trending towards younger individuals, becoming a major impediment to good vision [1]. At present, the chief operative intervention for myopia is corneal and intraocular refractive surgery, and the commonly used corneal refractive surgery entails irreversible cutting of corneal tissue and the corneal nerve. This predisposes to dry eye and other sequelae that may further impact postoperative visual quality [2]. Owing to rapid developments in related technology and materials, especially the hydrophilic and highly biocompatible phakic posterior chamber intraocular lens (STAAR Surgical, Lake Forest, CA, USA), surgical options for patients with average or high degrees of myopia are now more competitive [3] and subject to constant innovation [4]. However, selecting an appropriately sized implantable collamer lens (ICL) in advance of surgery is still problematic. Improper sizing leads to ICL rotation or positional change, limiting the vault ratio corrected by second operation up to 8% [5]. Because the postoperative vault (i.e., distance between anterior surface of natural lens and posterior ICL surface) is integral to ICL surgical success, addressing this issue is critical. High vault may lead to the closure of anterior chamber Angle (ACA), increased iris pigment dispersion, resulting in poor aqueous circulation and high intraocular pressure, pupillary block glaucoma. If the vault is too low after operation, resulting in lens opacification and cataract and other diseases. In addition, a recent retrospective study in Korean population developed a postoperative ICL vault prediction and lens-sizing model, and found that a least absolute shrinkage and selection operator (LASSO) model using the aqueous depth, ACA distance, and lens thickness (LT) showed the best results for postoperative ICL vault prediction [6].

Inspired by this, we analyzed ICL recipients treated for high-degree myopia in the Affiliated Xuzhou Municipal Hospital of Xuzhou Medicine University (Jiangsu, China), and assessed preoperative anatomic parameters of anterior chamber and lens characteristics. We also explored factors implicated in postoperative vault changes, and generated a predictive model for postoperative vaults in this setting. We hope our findings could provide some references for establishing a simple and perfect predictive model on postoperative vault to reduce the risk of relative complications of ICL.

## Methods

### Study design and population

We collected clinical data of 137 patients (272 eyes) with high-degree myopia, who were all recipients of V4c ICLs at our hospital between June 2020 and January 2021. Inclusion criteria were as follows: (1) anterior chamber depth (ACD) $\geq$2.8 mm; (2) stable refractive state (annual refractive change $\leq$0.5 D during 2 years prior to implantation); (3) corneal endothelial cell density >2000 cells/mm$^2$; (4) myopia $\geq$-6.0 D; and (5) astigmatism <6.0 D. The following were grounds for exclusion: (1) glaucoma, uveitis, diabetic retinopathy, pigment

dissemination, pseudoexfoliation syndrome, or other eye diseases; (2) autoimmune disorders, such as systemic lupus erythematosus; and (3) ongoing pregnancy or lactation.

Our protocol adhered to principles of the Helsinki Declaration set forth by the World Medical Association, requiring all subjects be informed of the purpose and risks involved in study participation and grant signed surgical consent. This research was approved by the ethics committee of Xvzhou Hospital and the approval number is xyyll [2022] 037. Written informed consent was obtained from the patients (or their parents or legal guardian in the case of children under 18) to publish this paper.

## Measured parameters

Horizontal ciliary sulcus-to-sulcus diameter (hSTS) and crystalline lens rise (CLR) were measured preoperatively by ultrasonic biomicroscopy (UBM SW-3200; Tianjin Suowei Electronic Technology Co Ltd, Tianjin, China). An anterior segment analyzer (Pentacam; Oculus Optikgeräte GmbH, Wetzlar, Germany) was used to measure horizontal white-to-white diameter (hWTW), ACD, ACA, anterior chamber volume (ACV), CLR, and vault (measured under the state of natural amplification of dark light) at postoperative Year1 and Month 1 respectively. Natural LT was determined by optical biometry (IOL Master; Carl Zeiss AG, Oberkochen, Germany).

## ICL operative procedure

The same surgeon implanted all ICLs, carefully checked results of patients' exams beforehand, and marked corneas prior to mydriasis. Pupillary dilatation was achieved by instilling tropicamide eye drops 2 hours in advance of surgery. Once surface anesthesia was attained (obucaine hydrochloride, 20 g/L), a viscoelastic agent was injected into anterior chamber via ancillary incision in corneal limbus. Anterior chamber entry was gained by separate incision (2.5 mm at temporal side; 8 mm along transparent cornea), using the Staar injector for ICL insertion. Upon confirming its full natural expansion, viscoelastic was again injected anterior to ICL and iridial engagement of the four footplates ensured, aligning axial plane according to preoperative design. All viscoelastic was then extruded and the incision tightened to restore normal intraocular pressure. Finally, a coating of tobramycin and dexamethasone ophthalmic ointment as applied to conjunctival sac, and anchored a transparent shield to protect the eye.

## UBM examination

In supine position, patients received surface anesthesia (obucaine hydrochloride, 20 g/L) for immersion bath ultrasonic probe examination. While holding the probe's focal plane tangential to iridial plane of the tested eye, its central position is adjustable on ultrasonic images. Transducer frequency was set at 50 MHz, exploration depth at 9 mm, and window depth at 9 mm. All measurements were acquired at center of cornea, recording averages of three consecutive determinations for hSTS and CLR values.

## Pentacam examination

The pentacam examination conducted using the pentacam 70100 (OCULUS Pentacam®, Germany). Patients were seated, placing lower jaws on the mandibular pad, with foreheads close to the frontal band. Gazes were fixed on visual targets in the blue light band at center of rotation axis, blinking 1–2 times to distribute tear films evenly. The examiner used an operating lever to focus in accordance with screen prompts. Measurements were obtained after flashing of blue light for 5 seconds, done in darkened conditions (natural pupillary state) with eyes

open and still. Only images of adequate quality by QS display were deemed acceptable. Three consecutive readings of CLR (i.e., horizontal distance from angle of iris and cornea to apex of anterior pole of lens) were generated, recording average values.

## Statistical analysis

Recorded measurements were tested for normality, expressing data of normal distribution as mean ± standard deviation (M±SD). For non-normal data distribution, median (interquartile range) was specified. Paired-sample t-test and Bland-Altman analysis were applied to CLR measurements (variably obtained), setting significance at $P<0.05$. The predictive model was formulated through multiple linear regression analyses of factors impacting postoperative vault, namely hSTS, CLR, hWTW, ACD, ACA, ACV, LT, and ICL size. The performance of the predictive model on vault after ICL was assessed via receiver operating characteristic (ROC) curve, area under the curve (AUC) and point of tangency (inspection level: $\alpha = 0.05$). All computations were driven by standard software (SPSS v25.0; IBM Corp, Armonk, NY, USA).

## Results

### Bland-Altman analysis

Mean age of the patients was 26.50±5.01 years (range, 19–37 years), and mean spherical power was -9.53±1.77 D (range, -6.25 to -15.75 D). The average CLR measured by UBM (0.561 ±0.134) was clearly lower than the corresponding value measured by Pentacam (0.683±0.196), indicating significant difference between methods by paired sample t-test (t = -4.901; $P<0.000$). Bland-Altman analysis revealed a good consistency in the two measurement methods and substantial correlation (r = 0.316; $P = 0.000$) (Fig 1).

### Multiple linear regression

Our prediction model relied on multiple linear regression analyses of factors impacting post-operative vault, including hSTS, CLR, hWTW, ACD, ACA, ACV, LT, and ICL size. At a

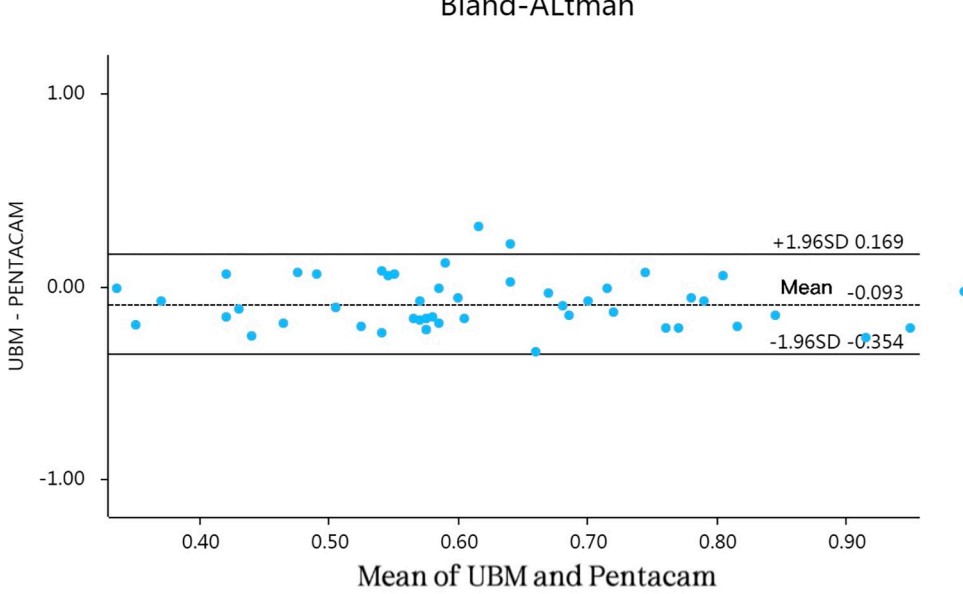

**Fig 1. Bland-Altman analyses of CLR values measured by UBM and Pentacam respectively.**

**Table 1. Multiple linear analyses of parameters impacting postoperative vault.**

| Variables | Standardized Coefficients (Year 1) | Standardized Coefficients (Month 1) | Tolerance | VIF |
|---|---|---|---|---|
| ICL size | 0.329 | 0.267 | 0.420 | 2.382 |
| SE, D | 0.035 | 0.101 | 0.719 | 1.390 |
| ACD, mm | 0.163 | 0.137 | 0.432 | 2.316 |
| ACA, ° | -0.036 | -0.049 | 0.665 | 1.503 |
| ACV, mm$^3$ | -0.020 | -0.018 | 0.507 | 1.972 |
| hWTW, mm | 0.531 | 0.611 | 0.326 | 3.071 |
| hSTS, mm | -0.720 | -0.783 | 0.359 | 2.783 |
| CLR, mm | -0.209 | -0.200 | 0.688 | 1.452 |

VIF, variance inflation factor; ICL, implantable collamer lens; SE, spherical equivalent; ACD, anterior chamber depth; ACA, anterior chamber angle; ACV, anterior chamber volume; hWTW, horizontal white-to-white diameter; hSTS, horizontal ciliary sulcus-to-sulcus diameter; CLR, crystalline lens rise.

variance inflation factor (VIF) <5, one may presume non-collinearity among various factors (Table 1). Model 1 (postoperative Year 1) = -2201.451 + 199.936 × ICL size + 4.185 × SE + 150.586 × ACD—2.022 × ACA—0.102 × ACV + 298.776 × WTW—291.828 × STS— 279.164 × CLR. Model 2 (postoperative Month 1) = -1389.987 + 131.843 × ICL size + 9.842 × SE + 102.793 × AVD—2.232 × ACA—0.075 × ACV + 278.768 × WTW— 257.632 × STS—216.633 × CLR.

## Predictive performance of the multivariate model

We drew ROC curves to assess the predictive value of the multivariate model (postoperative Year 1 and postoperative Month 1) on vault after ICL and determine appropriate points of tangency (Figs 2 and 3). The AUC of Model 1 was 0.847 (95% CI: 74.19–95.27), with an optimal threshold of 0.581 (sensitivity, 0.857; specificity, 0.724). For Model 2, the AUC value was 0.783 (95% CI: 64.94–91.64), with an optimal threshold of 0.522 (sensitivity, 0.917; specificity, 0.605). The results indicated that multivariate model have relatively good predictive value on vault after ICL in both short-term and long-term.

## Discussion

In the current study, we explored the vault effects of CLR and anterior chamber parameters in highly myopic patients receiving ICLs. Also, we established a multivariate model to predict the vault after ICL and assessed its predictive value in patients from our hospital. The results showed that the prediction model, which including hSTS, CLR, hWTW, ACD, ACA, ACV, LT, and ICL size, have relatively good performance on vault after ICL in both postoperative Month 1 and Year 1.

Although ACD and WTW are currently recommended by the manufacturer (STAAR) as determinants of ICL size, some sources have acknowledged that factors impacting postoperative vaults in this setting are far more extensive [7]. If values are borderline, the above two parameters may not be ideal [8]. A reasonably sized ICL is important to ensure a safe vault in the early postoperative phase. Vault excess may lead to closure of anterior chamber angle, increased iris pigment dispersion, poor aqueous circulation, high intraocular pressure, pupillary block glaucoma, and greater likelihood of halo. On the other hand, an inordinately low vault after surgery heightens the risk of friction between ICL and natural lens, promoting lens opacities and cataract formation [9, 10]. Gauging the postoperative vault and selecting the

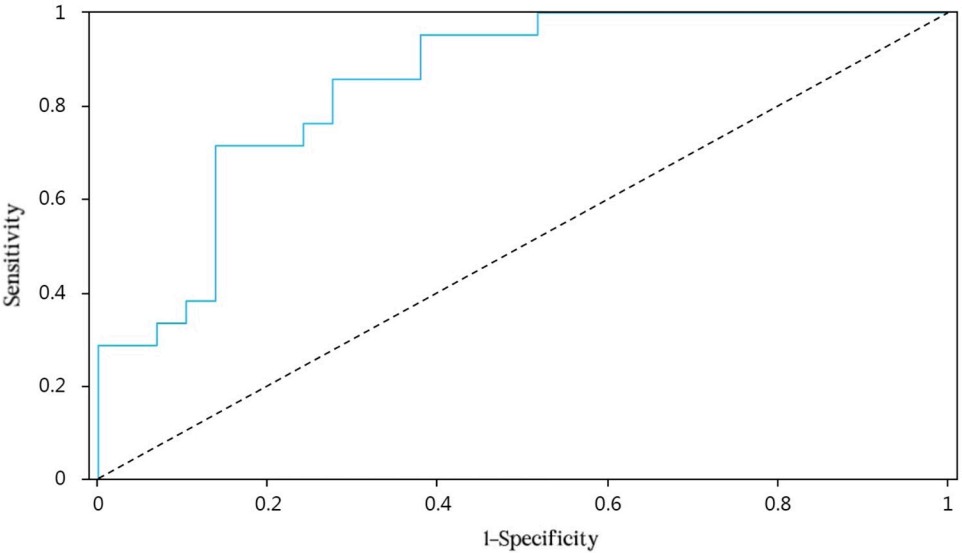

**Fig 2. The ROC curve of Model 1 in predicting vault estimates.**

most appropriately sized ICL is thus a challenging process, involving various aspects of eye anatomy, ICL geometry, and interaction between the two.

The postoperative vault is subsequently a pivotal element in safety of ICL placement, its ideal range being 250–750 μm. Too low vault may cause the natural lens capsule to become cloudy by friction, and however, Gonzalez-Lopez, et al. [11] considered that central port of the ICL may protect the patients from developing cataracts even with low vault. Herein, we used the conventional ICL placement, and the mean postoperative vault achieved at the end of observation was 587.08 ± 172.26 μm (95% CI: 546.60–627.56), reflecting ideal state. As Trancón et al. [12] have maintained, structural anatomy of the eye confers distinctive vault impact after surgery, perhaps through damping effects of landing points in ICL placement. However,

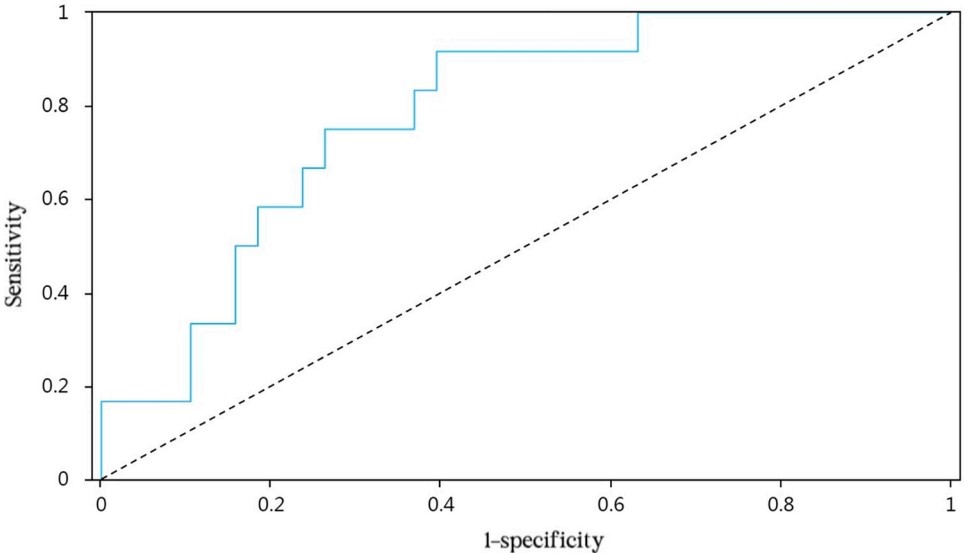

**Fig 3. The ROC curve of Model 2 in predicting vault estimates.**

their investigation incorporated none of our indices (hSTS, ACV, etc) as potential factors. In multiple linear regression analyses, we determined that both hSTS and ACV negatively correlated with postoperative vault.

The vault is also a parameter likely to vary in anterior or posterior axis of the eyeball. It is therefore reasonable to assume that anatomic variations and lens inconsistencies may impose vault fluctuations along the same axis. Although only a few clinical studies of CLR, CLR is an important parameter in preoperative patient assessments. Our findings show a negative correlation between CLR and postoperative arch height, which consistent with what previous studies discovered [13, 14]. Patients with higher CLRs tend to have lesser postoperative vaults; and given the geometric design of an intraocular lens, the inherent sagittal depth of an ICL is determined by its diopters and overall size. The higher the degree of myopia, the larger the diameter, and the greater the ICL sagittal depth appears to be. Because a lens is fixed in ciliary sulcus, STS may provide a more suitable basis for sizing purposes [15].

Various formulas utilizing UBM-acquired STS determinations have been advanced to aid in selecting lens size. In particular, Zheng et al. [16] have devised the following method for predicting postoperative vault: Vault (mm) = 1.785 + 0.017 × ACA + 0.051 × Lenscur—0.203 × WTW, where ACA is anterior chamber area and Lenscur is central curvature radius of anterior lens surface. Such efforts seemingly underscore the impact of anterior lens surface on lens size and vault after ICL placement, although clinical outcomes and postoperative vault calculations (via hSTS and hWTW) have proven similar in meta-analysis [5]. However, UBM measurements warrant sustained eye contact with a sensor probe and are troubled by examiner manipulation, device resolution concerns, or deviations in tested eyes. Inspector subjectivity may thereby skew measured values, and the repetition rate is apt to be poor. This is why UBM-based formulas are not in wide usage by ophthalmologists.

Pentacam is a non-invasive means of clear image-based assessment, with good repeatability. It is well-suited for structural imaging of anterior segment and is easily done, enabling intuitive and faster evaluations [17]. Consequently, we contend that anterior chamber parameters measured by Pentacam may be reliably used in calculating ICL size. Despite the consistency shown in CLR readings obtained by UBM and Pentacam (r = 0.316; $P$ = 0.000, Spearman's correlation), we relied solely on Pentacam-acquired CLR results for our predictive model, demonstrating that ICL size positively correlates with CLR (r = 0.253; $P$ = 0.032). Nakamura et al. [18] have also used multiple regression analyses in patients with neurotrophic keratitis, results for CLR (relative to ICL size) were similar to ours.

Another overlooked dynamic is that of compressive force. In differently sized implants, similar compression may produce disparate results. Otherwise stated, a higher compressive effect exerted in the largest of lenses may result in greater lens bulge and higher vaults. Actually, there are numerous ocular biologic parameters that impact the postoperative vault [19]. Present study findings confirm that CLR, reflecting lens lordosis, correlates negatively with postoperative vault but correlates positively with a value such as hWTW (r = 0.151; $P$ = 0.207), reflecting ocular transverse diameter. In borderline circumstances, a smaller-sized ICL may be selected, if CLR is low. If CLR is high, a larger CLR size is otherwise permissible. Ultimately, it is of utmost importance that the appropriate ICL size is selected prior to surgery.

The predictive model in this study was generated through multiple linear regression analyses, conducting the risk-scoring aspect by ROC curve evaluation. The AUC for vault prediction by Model 1 was 0.847 (95% CI: 74.19–95.27), with a threshold of 0.581 (sensitivity, 0.857; specificity, 0.724). In Model 2, respective values were 0.783 (95% CI: 64.94–91.64) and 0.522 (sensitivity, 0.917; specificity, 0.605), both models offering high predictive potential for postoperative vault. This risk-scoring approach is simple and effective, using the equation provided to calculate postoperative vault estimates (at two time points) in advance of surgery. Through

relatively objective measures, ICL size selection is facilitated upfront, and more personalized surgical plans are ensured for better clinical management.

This study does have several notable limitations. One is the follow-up period was only one year, and another is the matter of postoperative pupillary size, which may exert some vault impact [20]. Because our preoperative Pentacam determinations of anterior chamber parameters required darkroom conditions, and studies exploring vault effects of preoperative pupillary size have identified only weak correlation [21, 22], we disregarded pre- and postoperative pupillary size considerations. We measured the vault in the darkroom conditions, whereas previous researchers indicated that vault is dynamic and changes depending on light conditions [23], thus it may cause bias. In addition, the study population was in Chinese that limited the extrapolation of our study results. In the future, the external validation should be performed to develop and verify the regression model on predicting vault after ICL is needed.

In summary, CLR and various anterior chamber parameters are important determinants of postoperative vault after ICL placement. Using our predictive model, estimates of postoperative vault may be calculated before surgery, helping to guide ICL size selection and provide personalized treatment plans. In those patients with borderline values, unnecessary secondary operations may thus be avoided.

## Conclusion

CLR and anterior chamber parameters are important determinants of postoperative vault after ICL placement. A multivariate regression model may serve in large part as a predictive gauge, effectively avoid postoperative complication.

## Author Contributions

**Data curation:** Quan Zou, Sen Zhao.

**Formal analysis:** Quan Zou, Sen Zhao, Lei Cheng, Ping Yuan.

**Funding acquisition:** Ran Zhu.

**Investigation:** Quan Zou, Sen Zhao, Lei Cheng, Ping Yuan.

**Methodology:** Quan Zou, Lei Cheng, Chao Song, Ping Yuan.

**Project administration:** Ran Zhu.

**Software:** Quan Zou, Lei Cheng, Chao Song, Ping Yuan.

**Validation:** Quan Zou, Chao Song, Ping Yuan.

**Visualization:** Quan Zou, Chao Song.

**Writing – original draft:** Quan Zou.

**Writing – review & editing:** Quan Zou, Sen Zhao, Ran Zhu.

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
