## [Decision Letter · Decision Letter 0]

25 Sep 2023

PONE-D-23-24759Effects of crystalline lens rise and anterior chamber parameters on vault after implantable collamer lens placementPLOS ONE

Dear Dr. Zhu,

Thank you for submitting your manuscript to PLOS ONE. After careful consideration, we feel that it has merit but does not fully meet PLOS ONE’s publication criteria as it currently stands. Therefore, we invite you to submit a revised version of the manuscript that addresses the points raised during the review process.

ACADEMIC EDITOR:1. please provide representative images of the UBM and the pentacam that were used for data extraction.2. Is there a possibility that the measurements taken by the UBM may be different from those of the pentacam because of 1. the supine position of the patient in the UBM (gravity) and 2. the effect of pressure on the eye by the water bath? please discuss.3. please discuss, or at least hypothesise, the difference in measurements if a non-contact instrument was used for the measurement, namely anterior segment OCT.4. could this model be applied to eyes with low degrees of myopia, or to toric ICL corrections in which there is a significant astigmatism on the eye? please discuss.

We look forward to receiving your revised manuscript.

Kind regards,

Nader Hussien Lotfy Bayoumi, M.D., FRCS (Glasgow)

Academic Editor

PLOS ONE

Journal Requirements:

"This work was supported by Funding: Postgraduate Research & Practice Innovation Program of Jiangsu Province (SJCX22-1263)."

"NO authors have competing interests"

Reviewers' comments:

Reviewer's Responses to Questions

**Comments to the Author**

1. Is the manuscript technically sound, and do the data support the conclusions?

Reviewer #1: Yes

Reviewer #2: Yes

2. Has the statistical analysis been performed appropriately and rigorously? 

Reviewer #1: Yes

Reviewer #2: No

3. Have the authors made all data underlying the findings in their manuscript fully available?

Reviewer #1: Yes

Reviewer #2: Yes

4. Is the manuscript presented in an intelligible fashion and written in standard English?

Reviewer #1: No

Reviewer #2: Yes

5. Review Comments to the Author

Reviewer #1: The authors present a study about the importance of crystalline lens rise in postoperative vault of the ICL.

1. English grammar must be thoroughly revised by an English native speaker. Example (line 60): ‘ f the vault is too low after operation, resulting in lens opacification and cataract and other diseases.’ This sentence does not make sense.

2. Line 45. The author state that ‘the chief operative intervention for myopia is corneal and intraocular refractive surgery, which entails irreversible cutting of corneal tissue and the corneal nerve.’ There is no cutting of corneal tissue in intraocular refractive surgery.

3. Line 49. The authors state that ‘Owing to rapid developments in related technology and materials, especially the hydrophilic and highly biocompatible phakic posterior chamber intraocular lens (STAAR Surgical, Lake Forest, CA, USA), surgical options for patients with average or high degrees of myopia are now more competitive.’ However, the material of the ICL has been the same for more than 20 years.

4. Line 73. ‘corneal endothelial cell density > 2000 mm2.’ It should be 2000 cells/mm2.

5. Line 81. ‘Mean patient age was 26.50±5.01 years (range, 19-37 years), and mean spherical power was -9.53±1.77 D (range, -6.25 to -15.75 D). The data were accessed in Jun, 2022.’ This is not Methods; it s is Results.

6. Line 102. ‘using a special injector for ICL insertion.’ Do you mean the Staar injector?

7. The authors do not explain the light conditions in which vault was measured. According to the research of the group of Gonzalez-Lopez, vault is dynamic and changes depending on light conditions. This should be explained and discussed: Gonzalez-Lopez F, et al. Dynamic assessment of variations in pupil diameter using swept-source anterior segment optical coherence tomography after phakic collamer lens implantation. Eye Vis (Lond). 2021;8(1):39.

8. The authors state that vault below 250 microns is not safe. However, the central port of the ICL may protect the patients from developing cataracts even with low vault. Please read and discuss this paper: Gonzalez-Lopez F, et al. Long-term assessment of crystalline lens transparency in eyes implanted with a central-hole phakic collamer lens developing low postoperative vault. J Cataract Refract Surg. 2021;47(2):204-210.

9. Line 195. The authors state that ‘clinical studies of CLR are few’. Please read and discuss this paper: Gonzalez-Lopez F, et al. Determining the Potential Role of Crystalline Lens Rise in Vaulting in Posterior Chamber Phakic Collamer Lens Surgery for Correction of Myopia. J Refract Surg. 2019;35(3):177-183.

10. The authors do not mention as a limitation of the study the fact that they are studying only a Chinese population. Please read and discuss this paper: Lei Q, et al. Distribution of ocular biometric parameters and optimal model of anterior chamber depth regression in 28,709 adult cataract patients in China using swept-source optical biometry. BMC Ophthalmol. 2021;21(1):178.

Reviewer #2: This paper analyzed the postoperative vault and preoperative measurements for ICL surgery. The authors should summary the recent previous literature. There is also no comparative study design. There is no validation of the developed model at all.

1. Recent advances of this field are not included in the manuscript. Please read “Development of an implantable collamer lens sizing model: a retrospective study using ANTERION swept-source optical coherence tomography and a literature review”. Please review recent literature related to this study.

2. Please specify the version of pentacam. In my clinic, the pentacam HR device cannot measure CLR.

3. Please show the result images of pentacam and UBM. Please mark which variables were included in the final model.

4. What is the ground truth or labeled data for ROC curves?

5. The result section does not show the validation of the model. MAE or MedAE of the final model should be shown.

6. External validation should be performed. Training and testing partitioning is necessary for the regression model development.

7. Please show the comparison analysis with the formula of the manufacturer or other previous formulas.

6. PLOS authors have the option to publish the peer review history of their article (what does this mean?). If published, this will include your full peer review and any attached files.

Reviewer #1: No

Reviewer #2: No

---

## [Author Response · Author response to Decision Letter 0]

4 Dec 2023

Responses to Reviewer #1’s comments

The authors present a study about the importance of crystalline lens rise in postoperative vault of the ICL.

1. English grammar must be thoroughly revised by an English native speaker. Example (line 60): ‘f the vault is too low after operation, resulting in lens opacification and cataract and other diseases.’ This sentence does not make sense.

Response: Thank you for your valuable comments. We have checked and revised the grammatical mistake in the manuscript, and asked a native English-speaker for help.

2. Line 45. The author state that ‘the chief operative intervention for myopia is corneal and intraocular refractive surgery, which entails irreversible cutting of corneal tissue and the corneal nerve.’ There is no cutting of corneal tissue in intraocular refractive surgery.

Response: Thank you for your comment. To our knowledge, there are two main surgical interventions for myopia, including corneal refractive surgery and intraocular refractive surgery. The corneal refractive surgery requires irreversible cutting of corneal tissue and corneal nerves to change the curvature of the cornea to correct myopia. Intraocular refractive surgery corrects myopia by implanting an intraocular lens without cutting corneal tissue. In addition, we have revised the description on this sentence to make it easier to understand.

3. Line 49. The authors state that ‘Owing to rapid developments in related technology and materials, especially the hydrophilic and highly biocompatible phakic posterior chamber intraocular lens (STAAR Surgical, Lake Forest, CA, USA), surgical options for patients with average or high degrees of myopia are now more competitive.’ However, the material of the ICL has been the same for more than 20 years.

Response: Thank you for your valuable comment. Strampelli et al. designed the first non-folding intraocular lens with angular support in 1953. Barraquer et al. first reported the phakic intraocular lens (PIOL). However, due to the poor biocompatibility of the lens material, it may lead to a series of postoperative complications, such as corneal endothelial cell decompensation, poor aqueous circulation leading to glaucoma, iridocyclitis. In 1980, Fechner and Bikoff et al. improved the implantable intraocular lens, which made the intraocular lens more stable, but the predictability of postoperative results and complications were not significantly improved. These intraocular lenses are made of rigid polymethacrylic acid, which has been improved several times. In 1993, the Swiss STAAR company took the lead in the development of implantable artificial crystals (ICL), and subsequently developed a variety of models of artificial crystals such as V2, V3, V4, V4c, which greatly reduced the probability of complications such as vault obstruction and cataract during surgery.

4. Line 73. ‘corneal endothelial cell density > 2000 mm2.’ It should be 2000 cells/mm2. 

Response: We have revised this unit into “cells/mm2”

5. Line 81. ‘Mean patient age was 26.50±5.01 years (range, 19-37 years), and mean spherical power was -9.53±1.77 D (range, -6.25 to -15.75 D). The data were accessed in Jun, 2022.’ This is not Methods; it’s is Results.

Response: Thank you for your comment. We are agree with you and have moved these results into the Results section.

6. Line 102. ‘Using a special injector for ICL insertion.’ Do you mean the Staar injector?

Response: We actually mean the Staar injector, and have revised this sentence.

7. The authors do not explain the light conditions in which vault was measured. According to the research of the group of Gonzalez-Lopez, vault is dynamic and changes depending on light conditions. This should be explained and discussed: Gonzalez-Lopez F, et al. Dynamic assessment of variations in pupil diameter using swept-source anterior segment optical coherence tomography after phakic collamer lens implantation. Eye Vis (Lond). 2021;8(1):39.

Response: Thank you for your valuable comment. We have carefully read this article by Gonzalez-Lopez F, et al. In the current study, the examination was conducted in a dark room environment, and the vault was measured under the state of natural amplification of dark light. We have added the discussion on these conditions and the reference.

8. The authors state that vault below 250 microns is not safe. However, the central port of the ICL may protect the patients from developing cataracts even with low vault. Please read and discuss this paper: Gonzalez-Lopez F, et al. Long-term assessment of crystalline lens transparency in eyes implanted with a central-hole phakic collamer lens developing low postoperative vault. J Cataract Refract Surg. 2021;47(2):204-210.

Response: Thank you for your valuable comment. We have read this paper by Gonzalez-Lopez F, et al. In our opinion, the accepted ideal range of arch height is 250-750μm, too low vault may cause the natural lens capsule to become cloudy by friction. We have added the discussion on Gonzalez-Lopez’s findings in our manuscript.

9. Line 195. The authors state that ‘clinical studies of CLR are few’. Please read and discuss this paper: Gonzalez-Lopez F, et al. Determining the Potential Role of Crystalline Lens Rise in Vaulting in Posterior Chamber Phakic Collamer Lens Surgery for Correction of Myopia. J Refract Surg. 2019;35(3):177-183.

Response: Thank you for your valuable comment. We have added this paper in our manuscript.

10. The authors do not mention as a limitation of the study the fact that they are studying only a Chinese population. Please read and discuss this paper: Lei Q, et al. Distribution of ocular biometric parameters and optimal model of anterior chamber depth regression in 28,709 adult cataract patients in China using swept-source optical biometry. BMC Ophthalmol. 2021;21(1):178.

Response: Thank you for your comment. We are very agree with you that the study population was limited in the patients in China, and we have added this into the limitation.

Responses to Reviewer #2’s comments

This paper analyzed the postoperative vault and preoperative measurements for ICL surgery. The authors should summary the recent previous literature. There is also no comparative study design. There is no validation of the developed model at all.

1. Recent advances of this field are not included in the manuscript. Please read “Development of an implantable collamer lens sizing model: a retrospective study using ANTERION swept-source optical coherence tomography and a literature review”. Please review recent literature related to this study.

Response: We have read “Development of an implantable collamer lens sizing model: a retrospective study using ANTERION swept-source optical coherence tomography and a literature review”, and have added the recent advances of this field in the manuscript.

2. Please specify the version of pentacam. In my clinic, the pentacam HR device cannot measure CLR.

Response: Thank you for your valuable comment. We have added the version of pentacam. Besides, in our study, the CLR value was measured using UBM.

3. Please show the result images of pentacam and UBM. Please mark which variables were included in the final model.

Response: We have added the variables included in the final model in the Results section.

4. What is the ground truth or labeled data for ROC curves?

Response: The ROC curves was used to reflect the results on prediction and verification of the postoperative vault height of new patients using regression models.

5. The result section does not show the validation of the model. MAE or MedAE of the final model should be shown.

Response: Thank you for your valuable comment. In this study, we established a multivariate model for predicting the vault after ICL, including CLR as well as some anterior chamber parameters. We used the multiple linear regression analyses to select the variables included in the prediction model, and use ROC curves with AUCs to assess its predictive performance. We have not done the validation of the model using the MAE since we have not compared different prediction models via machine learning algorithm. In addition, we performed the multicollinearity test among the selected variables, and found no significant multicollinearity.

6. External validation should be performed. Training and testing partitioning is necessary for the regression model development.

Response: Thank you for your valuable comment. We are very agree with you that external validation should be performed. Therefore, in the future analyses, we will partition training and testing set in development of regression models basing on large and multi-center sample.

7. Please show the comparison analysis with the formula of the manufacturer or other previous formulas.

Response: Thank you for your valuable comment. Due to the study population limitation, we could not compared these results with other previous formulas. We think this suggestion is a very interesting research direction that we will explore in future studies if possible.

---

## [Editor Report · Decision Letter 1]

19 Dec 2023

Effects of crystalline lens rise and anterior chamber parameters on vault after implantable collamer lens placement

PONE-D-23-24759R1

Dear Dr. Zhu,

We’re pleased to inform you that your manuscript has been judged scientifically suitable for publication and will be formally accepted for publication once it meets all outstanding technical requirements.

Kind regards,

Nader Hussien Lotfy Bayoumi, M.D., FRCS (Glasgow)

Academic Editor

PLOS ONE
---

## [Editor Report · Acceptance letter]

11 Mar 2024

PONE-D-23-24759R1 

PLOS ONE

Dear Dr. Zhu, 

I'm pleased to inform you that your manuscript has been deemed suitable for publication in PLOS ONE. Congratulations! Your manuscript is now being handed over to our production team.

Kind regards, 

on behalf of

Professor Nader Hussien Lotfy Bayoumi 

Academic Editor

PLOS ONE